# Time-to-Treatment Initiation in a Decentralised Community-Care Model of Drug-Resistant Tuberculosis Management in the OR Tambo District Municipality of South Africa

**DOI:** 10.3390/ijerph20146423

**Published:** 2023-07-21

**Authors:** Joshua Oise Iruedo, Michael K. Pather

**Affiliations:** Division Family Medicine and Primary Care, Faculty of Medicine and Health Sciences, Stellenbosch University, Stellenbosch 7602, South Africa

**Keywords:** drug resistant tuberculosis (DR-TB), decentralised community DR-TB care model, OR Tambo district municipality, time-to-treatment initiation (TTTI), patient centred care (PCC)

## Abstract

Background: Drug-resistant tuberculosis (DR-TB) continues to challenge global efforts toward eradicating and having a tuberculosis-free world. Considering the high early mortality, especially among HIV-infected individuals, early diagnosis and prompt initiation of effective treatment are needed to significantly reduce mortality and halt transmission of DR-TB in the community. Aim: This study aims to assess the effectiveness of a community DR-TB care model with the specific objective of determining the Time-to-treatment initiation of DR-TB among patients in the OR Tambo district municipality. Methods: A prospective cohort study of patients with DR-TB was conducted in the OR Tambo district municipality of Eastern Cape Province, South Africa. Patients were enrolled as they presented for treatment initiation at the decentralised facilities following a diagnosis of DR-TB and compared with a centralised site. Results: A total of 454 DR-TB patients from six facilities between 2018 and 2020 were included in the analysis. The mean age was 37.54 (SD = 14.94) years. There were slightly more males (56.2%) than females (43.8%). Most of the patients were aged 18–44 years (67.5%), without income (82.3%). Results showed that slightly over thirteen percent (13.4%) of patients initiated treatment the same day they were diagnosed with DR-TB, while 36.3% were on the time-to-treatment target of being initiated within 5 days. However, about a quarter (25.8%) of patients failed to initiate treatment two weeks after diagnosis. Time-to-treatment initiation (TTTI) varied according to the decentralised sites, with progressive improvement with each successive year between 2018 and 2021. No demographic factor was significantly associated with TTTI. Conclusion: Despite rapid diagnosis, only 36% of patients were initiated on treatment promptly. Operational challenges remained, and services needed to be reorganised to maximise the exceptional potentials that a decentralised community DR-TB care model brings.

## 1. Introduction

Drug resistance poses a major threat towards eradicating tuberculosis (TB) by 2030. [1] Several patterns of drug resistance to anti-tuberculous drugs have emerged in the literature; rifampicin-resistant tuberculosis (RR-TB), multi-drug-resistant tuberculosis (MDR-TB), pre-extensively drug-resistant tuberculosis (pre-XDR TB), and extensively drug-resistant tuberculosis (XDR-TB) [1]. In individuals co-infected with human immunodeficiency virus (HIV), prompt diagnosis and early initiation of effective treatment rapidly diminishes the period of infectiousness of drug-resistant tuberculosis (DR-TB) and, thus, limits the spread of the disease [2,3,4].

The time-to-treatment initiation (TTTI) is defined as the time from diagnosis to commencement of DR-TB treatment. A very short time-to-treatment interval is critical to achieving sterile or non-infectious effluents to halt the disease’s onward transmission to significant others. Timeous treatment initiation is associated with a reduced time to culture conversion [4,5,6], and it is anticipated that prompt initiation of effective treatment will improve outcomes. An acceptable time recommended by the South African National Department of Health is five days [7]. The literature provides varying times from diagnosis to treatment depending on the setting and diagnostic modality used. Prior to the advent of molecular diagnostics for TB, a range of 14–19 days in South Africa [5,8,9] using mycobacterium growth indicator tube (MGIT) culture with phenotypic drug susceptibility testing (DST) has been reported. A range of 60–190 days was reported in China, where conventional (solid) culture and DST were used [10]. Similarly, the TTTI of 10–36 days was reported in South Africa with Xpert^R^ MTB/Rif assay, [9,11,12] all in a centralised model of care.

Evidence supports the hypothesis that primary DR-TB is the predominant form compared to acquired DR-TB [13,14]. As such, it is imperative that clinicians should focus on early diagnosis and initiation of effective treatment to mitigate the spread of DR-TB. With the advent of novel diagnostics, which have reduced the turn-around time for sputum Xpert^R^ MTB/RIF assay to two hours, it is possible to implement effective treatment promptly. An Xpert^R^ MTB/RIF result showing rifampicin resistance is immediately followed by the MDR-TB reflex tests, which include TB microscopy, MGIT culture, and line probe assay-1 (LPA1), which test for resistance against isoniazid and rifampicin only [15].

Before the 2018 decentralised community DR-TB care model in the OR Tambo district, individuals diagnosed with DR-TB were admitted to a specialised centre (Centralised model) in East London in the Buffalo City Metropole (BCM). Often, patients were taken further away (an average of 250–400 Km) from their families and support structures. As the number of patients increased, the specialised hospital became overwhelmed, leading to unnecessary delays in treatment initiation ranging from 18 to 64 days in the King Sabatha Dalindyebo (KSD) sub-district of the OR Tambo district municipality [9].

With the advocacy of a patient-centred care (PCC) approach to DR-TB management [16], many countries have adopted a decentralised strategy as it is deemed to be more cost-effective and has better outcomes than a centralised model. A study in South Africa reported the average cost per patient in a decentralised strategy as 3–4.5 times lower compared to other models of care [17] and with better outcomes [18]. The WHO [16] reported more loss to follow-up (LTFU), more deaths, and low treatment success among patients managed through the centralised model [16].

In the OR Tambo district, a decentralised community care model of DR-TB management was adopted in 2018. It was, therefore, necessary to evaluate the impact of new diagnostic technologies in the OR Tambo district municipality under decentralised community DR-TB care. In addition to implementing the decentralised community-care model of DR-TB management in 2011, [19] the South African government replaced injectable anti-tuberculous drugs with bedaquiline and linezolid for DR-TB. However, there is a paucity of information on the effectiveness of these strategies in the rural communities of the Eastern Cape. Therefore, this study examines the time-to-treatment initiation (TTTI) in patients diagnosed with DR-TB in the OR Tambo district municipality of the Eastern Cape Province, South Africa. In addition, the study reports on the proportion of patients with DR-TB who were initiated on treatment within five working days among the cohort following the national guideline [7].

## 2. Methods

### 2.1. Design and Setting

This prospective cohort study was conducted in all five sub-districts of the OR Tambo municipality of the Eastern Cape. The OR Tambo District is the most populous district municipality in the Eastern Cape Province, with a total population of 1,374,092 and a population density of 113.6 people per km^2^ [20]. It has five sub-districts, which include King Sabatha Dalindyebo (KSD), Nyandeni, Mhlontlo, Port St. Johns (PSJ), and Ingquza Hill. The residents depend mainly on the public sector for healthcare, and just less than 5% have access to medical schemes [20]. There are 136 clinics, ten primary healthcare clinics (PHCs), nine districts, two regional hospitals, a central hospital, and four private sector hospitals [20].

In 2015, the OR Tambo district had a higher TB incidence than the national average; 571 cases versus 520 cases per 100,000 populations per year. However, rifampicin-resistant tuberculosis (RR-TB) confirmed the client rate was slightly less than the national average, 5.4% versus 6.1%. Similarly, the MDR-TB treatment success rate of 39.2% was lower than the national target rate of 55% in 2013/2014 [20]. Except for people under five, HIV and TB remain the leading causes of death across all age categories, with non-communicable diseases accounting for significant mortality among the elderly.

### 2.2. Diagnosis of DR-TB

The OR Tambo district has four national primary healthcare laboratories equipped with TB diagnostic facilities (TB microscopy and Xpert MTB/Rif assay) located at Ingquza Hill, Nyandeni, Mqanduli (KSD), and a national tertiary laboratory located at the Nelson Mandela Academic Hospital complex which provides advance TB diagnostic services including TB microscopy, culture, Xpert MTB/Rif assay, and the line probe assay (LPA). There are also private laboratory services mainly affiliated with the private hospitals within the district. Patients presenting with a cough at any health facility are encouraged to produce sputum (National Tuberculosis Control Program), which is immediately sent to the nearest diagnostic laboratory for detection of mycobacterium tuberculosis. In other words, primary diagnosis is based on self–reporting. However, following the diagnosis of DR-TB, there is a screening of close contacts (active surveillance), usually household members and occupational contacts, by the ward-based outreach team (WBOT).

### 2.3. Standard of Care for DR-TB in the Study Setting

During the intensive phase of treatment, patients were initiated on treatment and reviewed monthly at a weekly clinic. At the same time, the mobile team, community health workers (CHWs), Hospice team, and the ward-based outreach team (WBOT) visited patient daily at home for daily injections in 2018/2019 before the adoption of all oral, injection-free regimen in the year 2019, and to enquire about adverse events. The patient’s weight is taken at each clinic visit, all outstanding blood and sputum results are reviewed, and adherence counselling is strengthened. During the continuation phase, patients were seen on a monthly basis for review. Audiometry was performed at baseline, and anytime the patient reported problems with hearing. Following the introduction of bedaquiline-containing oral regimens, the need for audiometry was reduced significantly, and electrocardiography (ECG) was performed regularly to monitor the QT interval.

Patients with pre-XDR and XDR-TB are usually admitted to the specialist (centralised) centre for treatment initiation and care until they are stabilised, following which they may be down-referred to their designated decentralised centres for follow-up.

### 2.4. Participants and Sample Size

All patients diagnosed with DR-TB between June 2018 and December 2020 in the OR Tambo district municipality were eligible for inclusion. According to the global TB report [1], South Africa had an estimated MDR-TB prevalence of 1.8% among newly diagnosed patients (primary) and 6.7% in individuals previously treated for TB. The sample size of 406 participants was estimated for a two-sample proportions test (Pearson’s Chi-squared test) with alpha = 0.05, power = 0.90, delta = 0.16, p1 = 0.39 and p2 = 0.55, with 203 patients per group using a purposive sampling method. Participants were included if they had received a microbiological diagnosis of DR-TB, residing in the OR Tambo district, and followed up at decentralised facilities (Mthatha gateway, St. Barnabas gateway, Zithulele gateway, Holy Cross gateway, Bambisana clinic) or centrally at Nkqubela hospital in Buffalo City Metropolitan municipality.

### 2.5. Study Procedure

All patients diagnosed with DR-TB undergo post-test counselling at the diagnosing facility (decentralised site), during which they are informed about their diagnosis. This is followed by pre-treatment counselling and an invitation to participate in the study. Information about the study, its purpose, and the questionnaire was provided to the participants verbally and through an information sheet at the initiation of treatment for DR-TB across the decentralised sites. Professional nurses were identified and trained from each decentralised site on the study purpose and process who recruited eligible participants and administered the questionnaire.

### 2.6. Data Collection

In addition to data collected at the enrolment of patients, more information was collected from the TB registers and patient’s case folders at the decentralised and centralised sites, the National Health Laboratory Services (NHLS) records, and the electronic drug resistant database, EDRWeb. During the fieldwork, patients with more than one case record were assessed as separate files only if the diagnosis and year of commencement of treatment were more than a year apart, as most patients were on a short regimen. The sequence of data collection is as follows:

STEP 1: Records of all patients from the OR Tambo district were reviewed at the diagnostic centres. Data collected included name, date of diagnosis, diagnostic modality, type of resistance, and sociodemographic details such as name, age, and sex. These were used to track the patient at the treatment sites.

STEP 2: At the treatment centres, patients were informed about the study, and an informed consent form was signed by willing participants, following which the patient’s level of income, occupation, education, HIV status, and comorbidities were obtained while the name, age, and sex were corroborated.

STEP 3: Additionally, at the treatment centres, the treatment commencement date, age, sex, and type of regimen, type of resistance, were collected from the case record and EDRWeb.

### 2.7. Outcome Measures

The study’s primary outcome measure is the time-to-treatment initiation (TTTI) of DR-TB. This is the time interval between diagnosis and commencement of DR-TB treatment. Mathematically, it is the difference between the date of treatment initiation and the date of diagnosis. Early or prompt treatment initiation is defined as the commencement of DR-TB treatment within five days of diagnosis [7]. The proportion of patients initiated within the recommended five days was disaggregated by clinic types, model of care, and diagnostic modalities.

### 2.8. Covariates

Sociodemographic and clinical covariates were included in the study. Age, sex, level of education, occupation, source of income, and social history (alcohol and substance use) were extracted from the medical records. The HIV status, diagnostic modalities, model of care, comorbidities, prior TB treatment, exposure environment (prison, mine, and health facility), notification, type of TB, and treatment regimen at the start and end were also extracted from the patients’ medical records.

### 2.9. Statistical Analysis

Data were coded in Microsoft Excel 2016 (Microsoft Corporation, Seattle, WA, USA) and analysed using STATA 13.1 (Stata Corp. LP, College Station, TX, USA). To analyse data that were not normally distributed, non-parametric statistics (median, interquartile range (IQR), the Wilcoxon sum rank test, and the Kruskal–Wallis test) were used. Categorical data were reported using proportions and the 95% confidence interval (95% CI) and compared using the two-sample test of proportions. Logistic regression (adjusted and unadjusted odds ratio) was corrected for confounders, and a *p*-value of <0.05 was considered significant.

### 2.10. Ethical Considerations

Ethics approval was granted by the Health Research Ethics Committee (HREC) of Stellenbosch University (HREC Reference #: S18/01/013). Permission to carry out the study was given by the Eastern Cape Department of Health, the OR Tambo District Municipality and the Buffalo City Metro (BCM). All patients who participated in the study received a copy of the participant information leaflet in English and the local language (isiXhosa) before they signed the informed consent form. Confidentiality of information was ensured during and after the study. The study followed the Helsinki Declaration and Good Clinical Practice Guidelines.

## 3. Results

A total of 454 DR-TB patients from six facilities between 2018 and 2020 were included in the analysis. The mean age was 37.54 (SD = 14.94) years. There were slightly more males (56.2%) than females (43.8%). Most of the patients were aged 18–44 years (67.5%) and had no source of income (82.3%). Only 5.7% of patients were less than 18 years. Over three-quarters of patients (75.8%) were unemployed, while a few of them had worked at the mines (6.0%) or spent time in prison (10.3%). Most patients were from district clinics (90.7%), and only a few were from centralised (specialist) centres (9.3%). All the patients were captured on the EDRWeb.

Table 1 details the patient’s demographics, and Table 2 the clinical characteristics, as well as the diagnostic modalities.

As shown in Table 2, only 15% of patients with DR-TB had other comorbidities. Hypertension, diabetes mellitus, epilepsy and allergies were the most common comorbidities in this study. About eighty-two percent (82.8%) of the cohort reported no identifiable comorbid condition. One-third of them had a history of substance use. Over half (50.5%) were new patients, and two-fifths (39.7%) had used the first-line anti-TB drug. About a third (32.6%) of TB patients returned to treatment after relapsing and 12.3% after being lost to follow-up. Nearly all (92.7%) patients were notified and diagnosed with pulmonary TB. Most patients (85.7%) were placed on a short regimen at the start of treatment. Similarly, most patients (77.3%) were on a short regimen at the end of their treatment.

Table 3 shows that polyresistance was seen in over half of patients (52.2%), while 46.1% had rifampicin monoresistance. Most patients with DR-TB (89.9%) were diagnosed with Xpert^R^ MTB/Rif assay, about seven (6.8%) percent by LPA, while others were either with culture or not stated. The smear was positive in 41.6% of the 426 patients with recorded smear tests. The culture was positive in 70.2% of patients. KatG mutation was almost twenty percent (19.6%) of the isoniazid resistance. While 85.7% of the participants had an electrocardiogram, only 35.7% had an audiogram.

Table 4 shows that all patients initiated care upon notification, but the time-to-treatment initiation varied according to the initiating health facility. The median time-to-treatment was 7 days (interquartile range = 12 days). The median time-to-treatment was the same in the centralised and decentralised clinics. However, Zithulele (median TTT = 3 days) and Bambisana/St. Elizabeth clinics (median TTT = 4 days) have far lower treatment initiation times than other clinics.

Figure 1 showed that slightly over thirteen percent (13.4%) of patients initiated treatment the same day while 36.3% were on time-to-treatment target of being commenced within 5 days. However, about a quarter (25.8%) of patients failed to initiate treatment two weeks after diagnosis.

Table 5 shows no demographic factor was associated with early treatment initiation (defined as starting treatment within five days of diagnosis). The proportion of patients who initiated treatment early increased from 28% in 2018 to 50% in 2021. Similarly, the unadjusted model showed that the odds of starting treatment early were higher with advancing years. However, in the adjusted model, while the direction of the effect persists, the effect size did not reach a statistically significant level.

Table 6 shows that the proportion of patients who initiated treatment early (within five days of diagnosis) in the centralised care model (42.86%) was slightly higher than in the decentralised care model (35.68%). The difference was not statistically significant. Likewise, there was no statistically significant difference in time-to-treatment initiation between the patients who initiated treatment in the centralised care model and in the decentralised care model in both the adjusted {2 [0.84, 4.76]} and unadjusted models {1.35 [0.71, 2.57]} with TTTI being 7 days in both settings. However, receiving TB care at the hospital was associated with early initiation of treatment. For example, the proportion of patients who started treatment early in Mthatha Gateway was 25.1% compared to 68.7% in Bambisana/St. Elizabeth clinic. The adjusted and unadjusted models confirm this result. Patients who received care at Bambisana/St. Elizabeth (*p*-value 0.01) and Zithulele clinics (*p*-value 0.001) were over five times more likely to start treatment early than those in Mthatha Gateway. No difference was observed in diagnostic modality and time-to-treatment initiation.

## 4. Discussions

This study provides insight into the impact of the decentralised DR-TB care model in the Eastern Cape, South Africa, by assessing the time-to-treatment initiation in confirmed patients.

The study finds a high preponderance of DR-TB among poor people, as evidenced by the high rate of unemployment, no identifiable source of income, ex-miners, and ex-prisoners. The finding of male predominance in the incidence of DR-TB in the study is not surprising, given that the exposure environment (prison and mine) is dominated by men [21,22,23,24,25]. In addition, it should be noted that men are more likely to default TB treatment, stemming from an urgent need to return to work or possibly related to alcohol or substance abuse, thus, increasing their risk for acquired resistance [26,27]. According to the World Health Organisation Report, 56% of the global DR-TB cases were males [28].

Interestingly, except for HIV, this study finds the majority of the cohort (82.8%) without any other comorbidity. This finding indicates the vulnerability of the populace towards contracting DR-TB and underscores the need to implement measures to prevent the spread of DR-TB at the population level. Early diagnosis and prompt treatment initiation are required to halt DR-TB transmission in the community [29]. HIV remains a major driver of TB in the OR Tambo District Municipality. Of the 447 patients with documented HIV status, 62.9% were HIV positive, with 134 (47.7%) patients having their viral load reported on the case record, of which 9% were virally suppressed, 6% had low-level viraemia, and 14% had virological failure necessitating a change of ART regimen. The high proportion of HIV in this DR-TB cohort is comparable to finding from another setting in South Africa, with 69% HIV positives in their cohort [30].

Though diagnosis had been relatively fast using molecular tests (Xpert^R^ MTB/Rif assay and LPA), only 36.3% of the cohort initiated treatment (within five working days) according to the recommendation by the World Health Organisation [1] and the National Department of Health [7]. The majority (63.7%) of the cohort had delays in treatment initiation between seven and fourteen days or more. Similar delays in treatment initiation in an era of modern diagnostics had been reported in South Africa [12] though the reason for such delays was not given [12]. An unpublished interview report (Iruedo and Pather, 2023) showed that long travel distances to care facilities and high travel costs might play a role in these delays. Often time, people had to borrow money from neighbours to cover transportation costs.

There were fifteen patients for whom the diagnostic modality could not be ascertained. It is probable that these are the few patients who were either diagnosed at the private laboratories whose results could not be found or declared missing following futile efforts at securing a reprint or were transferred in from other districts and provinces with referral letters only. However, their presence in the EDRWeb means they were confirmed cases of DR-TB whose results could not be objectively verified by the researchers at the time. This may explain the ridiculously high odds ratio (14.82) among patients with missing results seen in Table 6.

All patients diagnosed with DR-TB were initiated on treatment in this cohort. This aligns with the WHO report of the latest trend between 2020 and 2022, where nearly all people diagnosed with DR-TB were enrolled on treatment [31]. This is in contrast to the period between 2015 and 2019, where treatment enrolment slightly lagged behind the diagnosis of DR-TB [31] owing to delays and loss to follow-up [12] or to high early mortality [32].

One pertinent finding from this study is the progressive improvement in Time-to-treatment initiation with each successive year; 2018 (TTTI = 8 days), 2019 (TTTI = 7 days), 2020 (TTTI = 7 days), and 2021 (TTTI = 5.5 days). This change may be partly explained by improvement in the DR-TB programme, including staff training, availability of more effective drugs, sending prompts and reminders for appointments (calls and messages), socioeconomic incentives (financial and food package), and a host of others. Support and incentives for patients with DR-TB have been shown to improve adherence [33,34] and health-related quality of life [35]. Additionally, this could reflect a change in the patient’s health-seeking behaviour due to the awareness created by all stakeholders involved with DR-TB management over the years or a natural survival instinct due in part to fear of the devastating impact of the COVID-19 pandemic. Future study should look into the actual reason for this observation.

Before 2018, DR-TB management was centralised mainly at the Buffalo City Metro (BCM), with patients admitted to facilities over 240km away from home. As the number of patients diagnosed with DR-TB increased, there were delays in treatment initiation mainly due to the unavailability of beds at the centralised hospital, personal arrangements as patients needed to be prepared for long-term separation from family, and transportation challenges. Implementing a decentralised community DR-TB care model in the OR Tambo district municipality has significantly minimised these delays. Most patients now commenced treatment within 7 days (with over 36% being initiated within 5 days), representing a significant improvement compared to 18 days using Xpert^R^ MTB/Rif assay and 29 days using LPA before decentralisation within the same district municipality [9]. This is comparable to a median time-to-treatment of 6 days obtained in Bangladesh, where the decentralised care model was first implemented on a large scale [4]. Another study in South Africa reported a higher proportion of patients (64%) initiating treatment early following diagnosis with Xpert^R^ MTB/Rif assay [36].

Of note is the use of LPA in the primary diagnosis of DR-TB in the district instead of the Xpert MTB/Rif assay. This could be explained by the LPA performed directly (direct LPA) on smear-positive sputum [37,38] and avoided a delay in diagnosis even before MGIT culture results were available. Furthermore, the inclusion of the LPA services package to the TB diagnostics at the National Health Laboratory Services (NHLS) in the Nelson Mandela Academic Hospital complex resident within the OR Tambo district municipality may have contributed to this observation, compared to when the LPA was performed in more distant East London (~226 km) or Port Elizabeth (~483 km).

One crucial finding was the difference in treatment initiation time per clinic or decentralised site. We see patients managed at the Zithulele Hospital and Bambisana/St. Elizabeth Hospital was initiated on target of three and four days as against eight days and 9.5 days at the Mthatha Gateway and St. Barnabas Gateway Clinics. What could explain this observation is the slightly different *modus operandi* of these clinics compared to the Zithulele and Bambisana hospital sites. Firstly, Mthatha Gateway Clinic is not within the Mthatha Hospital Complex and does not primarily diagnose DR-TB though it manages over 41% of all patients diagnosed with DR-TB in the OR Tambo district municipality with feeds from all the clinics in KSD and Mhlontlo sub-Districts. This accounted for the slight delay in presentation to the Mthatha Gateway decentralised site as patients first had to collect results at the diagnosing clinics before presenting to the Mthatha Gateway Clinic for TB management.

Most of the patients were diagnosed using Xpert^R^ MTB/Rif assay (89.9%) with direct read-off of rifampicin resistance, while 6.7% were diagnosed using the line probe assay (LPA). The high Xpert^R^ MTB/Rif assay uptake is commendable, as recommended by the WHO, to remove the hurdle of diagnostic delays and improve prompt treatment initiation [1]. The use of Xpert^R^ MTB/Rif assay and decentralisation have been reported as facilitators of DR-TB care, preventing early loss to follow-up (LTFU) and deaths due to lengthy diagnostic and treatment delays [39]. Prompt diagnosis and treatment are also associated with better outcomes [40], but in an earlier study in Johannesburg, South Africa, Xpert^R^ MTB/Rif assay diagnosis was associated with early time-to-treatment initiation but not treatment outcome [41].

In terms of the care received by the patients, a lesser number of patients had audiometry (35.7%), while the majority had ECG (85.7%) and blood (83.3%) investigations done. The low uptake of audiometry was related to the type of regimen used to manage the patients. Most patients had bedaquiline-containing injection-free regimens following the discontinuation of aminoglycosides in 2019 in the OR Tambo district municipality. Aminoglycosides (amikacin, kanamycin, or capreomycin) were the primary drugs responsible for irreversible ototoxicity [42]. A review reported that aminoglycosides were responsible for numerous cases of preventable hearing loss among patients managed for DR-TB exceeding fifty thousand cases annually [43]. Meanwhile, the high uptake of ECG among the cohort was due to the use of the bedaquiline-based regimen, known for its QT-interval prolongation and propensity for arrhythmias. Other medications, such as linezolid and fluoroquinolones may also cause QT-interval prolongation [44,45].

Poly-resistance accounted for 52.2%, while 45.6% were mono-resistant TB (Rif-mono or INH-mono resistance). Pulmonary TB (98.7%) was the predominant presentation, and extra-pulmonary TB accounted for only 1.3% of all cases. This had major implications in terms of the mode of spread of DR-TB among the affected communities. Among the cohort, rifampicin-resistant (RR-TB) tuberculosis (46.1%) was the predominant type, followed by MDR-TB (43.6%). Pre-XDR TB (5.2%) and XDR-TB (3.8%) were less represented. Most (90%) were managed as MDR-TB in the community in line with the policy guideline and WHO recommendations [46,47]. In contrast, pre-XDR and XDR-TB were addressed at the centralised (Nkqubela) TB Hospital in the Buffalo City Metropole.

A total of 82.6% were commenced on the short regimen, of which 5.3% had to be changed to the long regimen either due to a change in mutation, additional resistance against the baseline, or the development of comorbidity. All the patients diagnosed with rifampicin resistance (RR) using Xpert^R^ MTB/Rif assay had the MDRTB reflex test to confirm further resistance beyond rifampicin. This confirmatory drug susceptibility testing (DST) had particular relevance in formulating preventive therapy among household contacts of an index DR-TB patient. A study [48] reported the benefits of an expanded DST beyond Xpert^R^ MTB/Rif assay for similar purposes [48].

On resistance to first-line anti-TB medications, KatG mutation was the most common (14.5%), followed by a combined inhA and KatG mutation (5.1%) and inhA mutation (5.5%). rpoB gene mutation, which confers rifampicin resistance [49,50], is read directly from the Xpert MTB/Rif assay test, accounting for 43% of this study cohort and confirmed using the MDRTB reflex test. The overall KatG mutation of 20% has the chemotherapeutic implication of using high-dose isoniazid (15 mg/kg) in managing DR-TB, which is rendered ineffective by this high-level isoniazid resistance [51]. The WHO [52] recommended avoiding isoniazid when KatG mutation is present. Therefore, mutations play a role in formulating regimens when initiating patients on DR-TB treatment or preventive therapy.

## 5. Strengths and Limitations

The findings highlight the current practice and implementation gaps in DR-TB management in the OR Tambo district. In addition, the findings serve as a reference guide for future studies on time-to-treatment initiation in the Eastern Cape Province. Nonetheless, the limitations of the study cannot be ignored. This study was conducted in one of six districts of the province, and as such, findings may not necessarily reflect the general practice in the other districts. In addition, the relatively small number of patients managed at the centralised site (42{9.3%} versus 412{90.7%}) compared to the decentralised sites masked the true effect size of the difference between the two models of care. The COVID-19 lockdown in 2020 may have imparted the study outcomes negatively due to movement restrictions and business closures.

Future studies should examine the time to diagnosis of DR-TB and potentially differentiate primary from acquired DR-TB in the region.

## 6. Conclusions

The time-to-treatment initiation has significantly improved under decentralisation compared to an era of solely centralised care. However, with just over 36% of the patients initiating treatment within five days, it may be necessary to strengthen programmes to cater for patients over the weekend and reorganise services to prevent unnecessary delays.

## 7. Recommendation

The full benefits of rapid diagnostic technologies may not be realised if contextual factors (health systems delays) to starting treatment for people who are diagnosed with DR-TB are not addressed. Expanding access to DR-TB care by increasing the number of decentralised sites equipped with testing services in the district will reduce unnecessary delays in treatment initiation. This may also be cost-saving for the patients and move the district closer to meeting the UN goal of reducing the catastrophic cost of healthcare as we strive toward universal health coverage.

Therefore, as more people are having decentralised care (91% in this study), there is a need to:I.Decentralise the WHO rapid diagnostics, e.g., Xpert MTB/Rif, including the Xpert MTB/XDR testing, to peripheral and decentralised facilities. This may shorten the time from diagnosis to treatment initiation as the Xpert MTB/XDR assay offers DSTs for isoniazid, rifampicin, fluoroquinolones and second-line injectables. This assay can be deployed to decentralised sites to provide faster, near-patient access to second-line DSTs.II.Have more required resources (human and equipment) to effectively manage and monitor adverse events of DR-TB at decentralised levels. This should be complemented with mechanisms for early referrals to specialist care at central facilities.

## Figures and Tables

**Figure 1 ijerph-20-06423-f001:**
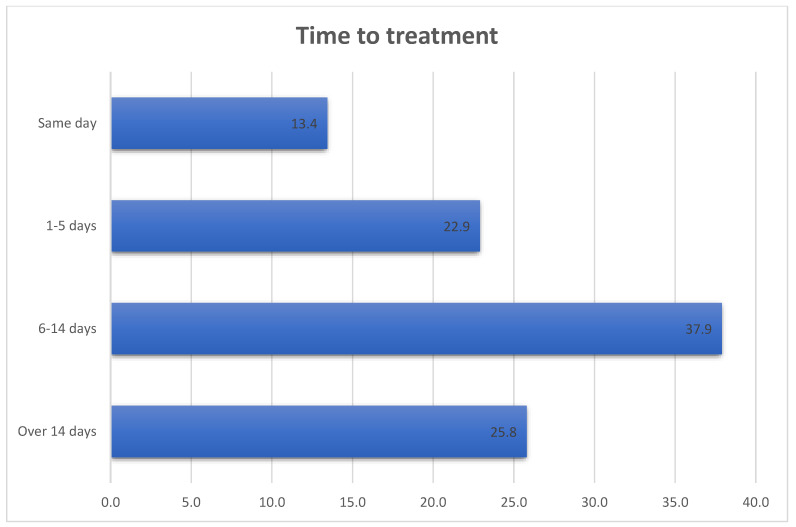
Time-to-treatment.

**Table 1 ijerph-20-06423-t001:** Patients’ demographic characteristics.

Variables	Frequency	Percent
Age (N = 454)		
1–17	26	5.7
18–24	57	12.6
25–34	128	28.2
35–44	121	26.7
45–54	53	11.7
55–86	69	15.2
Gender (N = 454)		
Female	199	43.8
Male	255	56.2
Education (N = 453)		
No education	91	20.1
Primary	104	23.0
Secondary	217	47.9
Tertiary	41	9.1
Income (N = 434)		
Salary	38	8.8
Casual	2	0.5
UIF	2	0.5
Grant	28	6.5
No income	357	82.3
Self-employed	7	1.6
Occupation (N = 433)		
Unemployed	328	75.8
Student	35	8.1
Pensioner	20	4.6
Grant	8	1.9
Government department	15	3.5
Private sector	19	4.4
Minor	6	1.4
Prisoner	2	0.5
Model of care (N = 454)		
Decentralised (District)	412	90.7
Centralised	42	9.3
Exposure Environment (N = 436)		
Prison	45	10.3
Mines	26	6.0
HCW	1	0.2
Both Prison and Mine	16	3.7
None	348	79.8

Missing for Education (1); No Income (20); Some Income2 (20); Occupation (21); Patient work (18).

**Table 2 ijerph-20-06423-t002:** Patients’ Clinical Characteristics.

Variables	Frequency	Percent
Comorbidities (N= 68)		
HTN	21	4.6
Type2DM	11	2.4
Epilepsy	10	2.2
Mental	3	0.7
Hearing	26	5.7
Allergies	2	0.4
Asthma	1	0.2
Clinic Names (N = 454)		
Mthatha Gateway (KSD)	187	41.2
Holy Cross (Inquza Hill)	63	13.9
Barnabas Gateway (Nyandeni)	60	13.2
Zithulele (KSD)	85	18.7
Bambisana (PSJ)	17	3.7
Nkqubela Chest Hospital (BCM)	42	9.3
Social History—Cigarette (N = 430)		
Users	143	33.3
Non users	287	66.7
Previous drug history (N = 448)		
New	226	50.5
Previously treated with 1st line drug	178	39.7
Previously treated with 2nd line drug	43	9.6
Unknown	1	0.2
Patient Category (N = 448)		
New	231	51.6
Relapse	146	32.6
After Loss to follow up	55	12.3
After Failure 1st line	13	2.9
After Failure 2nd line	3	0.7
Notification (N = 450)		
No	29	6.4
Yes	421	92.7
Type of TB (N = 452)		
Extra-pulmonary TB	6	1.3
Pulmonary TB	446	98.7
Type of regimen at the start of treatment (N = 443)		
Long	68	15.0
Short	375	82.6
Type of regimen at end of treatment (N = 436)		
Long	85	18.7
Short	351	77.3
HIV status (N = 447)		
Negative	165	36.9
Positive	281	62.9

NB: number of missing per variable: Clinic name (1); Comorbidities (10); HTN (10); Type2DM (10); Type1DM (10); Kidney (10); Cancer (10); Epilepsy (10); Mental (10); Liver (10); Hearing (10); Allergies (10); Asthma (10); Social History (24); Drug History (6); Patient Category (6); Notification (4); Type of TB (2).

**Table 3 ijerph-20-06423-t003:** Results of investigations.

Type of Resistance (N = 444)	Frequency	Percent
Poly	237	52.2
Mono	207	45.6
Type DR-TB (N = 445)		
Rifampicin Resistant	205	46.1
MDR	194	43.6
Pre-XDR	23	5.2
XDR	17	3.8
Isoniazid Resistance	6	1.4
Diagnostic Modality (N = 439)		
LPA	31	6.8
Xpert (GXP)	408	89.9
Smear Results (N = 426)		
Negative	237	52.2
Positive	189	41.6
Culture Results (N = 399)		
Negative	110	27.6
Positive	280	70.2
Contaminated	9	2.3
DST1 (N = 407)		
RR	175	43.0
INH and RR	218	53.6
INH	11	2.7
Sensitive	3	0.7
DST2 (N = 38)		
Fluoroquinolone resistance	8	1.8
Injectable resistance	12	2.6
Fluoroquinolone and Injectable resistance	18	4.0
LPA1 (N = 207)		
InhA (low-level isoniazid mutation)	25	5.5
KatG (high-level isoniazid mutation)	66	14.5
InhA and KatG (Combined mutations)	23	5.1
No Mutation	93	20.5
LPA2 (N = 181)		
gyrA/gyrB (Fluoroquinolone mutation)	8	1.8
rrs/eis (Aminoglycoside injectable mutation)	12	2.6
gyrA/gyrB and rrs/eis (Combined mutations)	18	4.0
No Mutation	143	31.5
Viral Load (N = 134)		
Suppressed	41	9.0
Low-level Viraemia	29	6.4
Virological Failure	64	14.1
CPT (N = 120) (cotrimoxazole prophylactic therapy)		
Yes	107	23.6
No	13	2.9
Baseline Investigations Blood (N = 453)		
No	75	16.5
Yes	378	83.3
Audiometry (N = 437)		
No	275	60.6
Yes	162	35.7
ECG (N = 431)		
No	42	9.3
Yes	389	85.7

NB: number of missing per variable: Type of Resistance (10); Type DR-TB (9); Diagnostic Modality (15); Smear Results (28); Culture Results (55); DST1 (47); DST2 (416); LPA1 (247); LPA2 (273); Viral load (320); Baseline Investigations Blood (1); Audiometry (17); ECG (23); Type of regimen at start of treatment (11); Type of regimen at end of treatment (18).

**Table 4 ijerph-20-06423-t004:** Distribution of Time-to-treatment.

Variables	Median Time-to-Treatment in Days	Interquartile Range
Overall (n = 454)	7	3–15
Years		
2018 (n = 149)	8	4–14
2019 (n = 161)	7	3–15
2020 (n = 128)	7	2–15
2021 (n = 16)	5.5	3.5–8
Clinic types		
District clinics (n = 412)	7	3–15
Centralised (n = 42)	7	3–15
Clinic Names		
Mthatha Gateway (n = 187)	8	5–22
Holy Cross (n = 63)	7	3–13
Barnabas Gateway (n = 60)	9.5	6.5–18
Zithulele (n = 85)	3	1–7
Bambisana/St. Elizabeth (n = 17)	4	1–6
Nkqubela Chest Hospital (n = 42)	7	3–15
Diagnostic Modality		
LPA (n = 31)	8	0–34
Xpert (n = 408)	7	4–15
Missing (n = 15)	0	0–0
Culture Results		
Negative (n = 110)	7	3–16
Positive (n = 280)	7	3–15
Contaminated (n = 9)	8	1–21
Missing (n = 55)	6	4–13

**Table 5 ijerph-20-06423-t005:** Binary logistic regression models showing the association between demographic factors and early treatment initiation.

Variables	Yes	UOR [95% CI] Unadjusted OR	AOR [95% CI] Adjusted OR
	N (%)		
Age			
1–24	36 (43.37)	Ref	Ref
25–34	40 (31.25)	0.59 [0.33, 1.05]	0.65 [0.36, 1.17]
35–44	45 (37.19)	0.77 [0.44, 1.37]	0.8 [0.45, 1.44]
Above 44	44 (36.07)	0.74 [0.42, 1.30]	0.7 [0.38, 1.28]
Gender			
Female	79 (39.7)	Ref	Ref
Male	86 (33.73)	0.77 [0.53, 1.14]	0.79 [0.53, 1.18]
Education			
No Education	37 (40.66)	Ref	Ref
Primary	38 (36.54)	0.84 [0.47, 1.50]	0.91 [0.50, 1.64]
Secondary	76 (35.02)	0.79 [0.48, 1.30]	0.78 [0.45, 1.34]
Tertiary	13 (31.71)	0.68 [0.31, 1.48]	0.71 [0.31, 1.59]
Missing	1 (100)	1 [1.00, 1.00]	1 [1.00, 1.00]
Years			
2018	43 (28.86)	Ref	Ref
2019	61 (37.89)	1.5 [0.93, 2.42]	1.43 [0.88, 2.32]
2020	53 (41.41)	1.74 * [1.06, 2.87]	1.62 [0.98, 2.69]
2021	8 (50)	2.47 [0.87, 6.99]	2.51 [0.87, 7.23]

* Significant odds ratio.

**Table 6 ijerph-20-06423-t006:** Binary logistic regression models show the association between clinic factors, diagnostic modalities, and early treatment initiation.

Variables	Yes	UOR [95% CI]	AOR [95% CI]
	N (%)		
Clinic types			
Decentralised (District)	147 (35.68)	Ref	Ref
Centralised	18 (42.86)	1.35 [0.71, 2.57]	2 [0.84, 4.76]
Clinics			
Mthatha Gateway	47 (25.13)	Ref	Ref
Holy Cross	22 (34.92)	1.6 [0.86, 2.95]	1.43 [0.75, 2.73]
Barnabas Gateway	13 (21.67)	0.82 [0.41, 1.65]	0.74 [0.35, 1.57]
Zithulele	54 (63.53)	5.19 *** [2.99, 9.01]	5.43 *** [3.07, 9.62]
Bambisana/St. Elizabeth	11 (64.71)	5.46 ** [1.91, 15.58]	5.05 ** [1.69, 15.11]
Nkqubela Chest(Centralised)	18 (42.86)	2.23 * [1.12, 4.48]	1 [1.00, 1.00]
Diagnostic Modality			
LPA	15 (48.39)	Ref	Ref
Xpert	136 (33.33)	0.53 [0.26, 1.11]	0.51 [0.23, 1.13]
Missing	14 (93.33)	14.93 * [1.74, 127.89]	14.82 * [1.60, 136.92]
Culture results			
Negative	33 (30)	Ref	Ref
Positive	105 (37.5)	1.4 [0.87, 2.25]	1.35 [0.79, 2.29]
Contaminated	4 (44.44)	1.87 [0.47, 7.39]	2.9 [0.69, 12.27]
Missing	23 (41.82)	1.68 [0.86, 3.29]	1.59 [0.73, 3.48]

* *p*-values < 0.05, ** *p*-values < 0.01, *** *p*-values < 0.001, UOR: unadjusted odds ratio, AOR: adjusted odds ratio.

## Data Availability

All data forms are kept safe and can be made available upon request. And data could be assessed at the respective service providing institutions with permis-sion by the Eastern Cape Department of Health, Bisho.

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
