# Peer review of "Time-to-Treatment Initiation in a Decentralised Community-Care Model of Drug-Resistant Tuberculosis Management in the OR Tambo District Municipality of South Africa"

_ijerph, 2023, doi:10.3390/ijerph20146423_

Round 1
Reviewer 1 Report
The study by Iruedo and Pather highlights an important public health topic in OR Tambo district of South Africa. Timely initiation of people with DR-TB on effective treatment is crucial to reduce community transmission of TB and to improve treatment outcomes. While there are some important findings, the study could benefit from some improvements especially in the study setting (diagnosis of DR-TB, treatment of DR-TB and study procedures). This is important to contextualize the results of the study and to inform both the discussion and the recommendations.
Major comments
The discussion and the recommendation could benefit from a detailed description of the study setting. While an attempt was made by the authors to describe the Standard of care for people with DR-TB and the study procedures, more detail needs to be added.
Diagnosis and management of DR-TB
If possible, state the diagnostic algorithm that was in use. If LPA is done, is it done on sputum positive deposits or culture deposits? Given that 62% were co-infected with HIV in this study, it might be possible that some people living with HIV produced pauci-bacillary specimens, making preliminary culture a requirement in order to produce visible bands on LPA. However, the authors make reference to use of sputum deposits for LPA in the discussion lines xx.
· How are results communicated to patients? (SMSs, calls or they visit the diagnostic facility)
· Are patients referred to either decentralized or centralized facilities to start treatment?
· Are there DR-TB concillia/committees that have to meet and discuss treatment options for patients?
Treatment for DR-TB
Are patients initially started on standardized treatment pending second line DST results to inform individualised treatment? Which group of patients are started on standardized therapy?
Study procedures
May the author chronologically show how data collection unfolded. Perhaps the authors may state that all records of people who were diagnosed with DR-TB were reviewed at diagnostic centers. Data that were collected included date diagnosis, results and sociodemographic details (e.g. names, age and sex) that were used to track patients at decentralized centers and centralized centers. In the next step, they can spell out data that were collected at treatment centres either i) directly from patients (income, education, occupation etc) or ii) from Case report forms and EDRweb (date of starting treatment; age, sex, etc). I would like to believe the dates were collected from records NOT from self-reports by the patients.
Minor comments
Outcome variable: state that it was difference between date of starting treatment and date of diagnosis.
Lines 322-324: Of note is the finding of no significant difference in treatment initiation time when LPA was used as a diagnostic modality as compared to Xpert in this cohort. This could be explained by the LPA done directly (direct LPA) on smear positive sputum (35, 36) even before MGIT culture results were available.
Since the study focused on interval between diagnosis and starting treatment, diagnostic modality may not have an effect on time to treatment initiation. Differences could have been observed had the study looked at interval from presumption of TB to diagnosis and treatment initiation.
Lines132-133: Study mentions that part of the inclusion criteria was microbiological diagnosis of DR-TB but in Table 4, there are 15 patients with missing diagnostic modality.
Define the acronyms used in the table in the legend of Tables.
Recommendation
To me one of the key recommendation is that as more and more people are having decentralized treatment, there is need to:
i) decentralize WHO rapid diagnostics e.g. Xpert MTB/Rif including the Xpert MTB/XDR testing to peripheral/decentralized facilities. This may shorten time from diagnosis to treatment initiation. The Gene Xpert MTB/XDR offers DSTs for isoniazid, rifampicin, FQs and second-line injectables. This assay can be deployed to peripheral areas to provide faster, near patient access to second line DSTs.
ii) As more people (91% in this study) are now in decentralized DR-TB care, more resources (human and equipment) are required to effectively manage and monitor adverse events of DR-TB at decentralized levels. This should be complemented with mechanisms for early referrals to specialist care at central facilities.
iii) The full benefits of rapid diagnostic technologies may not be realized if contextual factors (health systems delays) to starting treatment for people who are diagnosed with DR-TB are not addressed. Reasons for differences in TTTI in the various health facilities. Is it because of failure to adhere to TB guidelines in some facilities?
It would have been great had the authors given us an idea of the % of people who were switched to effective regimens after CDST results. This is crucial within the context of FQ resistance reported in this study. While shorter time to treatment is important, time to effective treatment is even more important to reduce the time when people are on suboptimal therapies. For this reason, the authors may recommend Xpert XDR tests to be deployed.
Tables: The tables may need a column for the total number of patients in each category. That way it will be easier to deduce how the row percentages were calculated.
Table 3: Check the denominator used to calculate the % of people who attained viral suppression. It seems the denominator used was all patients, including those who were HIV negative.
Line 49: Make it clear that successful outcomes depend on starting effective treatment regimens.
References: Some parts of the manuscripts are referenced using the APA while others are in Vancouver.
Minor edits of English language are required.
Author Response
RESPONSE TO REVIEWER No 1
Comments and Suggestions for Authors
The study by Iruedo and Pather highlights an important public health topic in OR Tambo district of South Africa. Timely initiation of people with DR-TB on effective treatment is crucial to reduce community transmission of TB and to improve treatment outcomes. While there are some important findings, the study could benefit from some improvements especially in the study setting (diagnosis of DR-TB, treatment of DR-TB and study procedures). This is important to contextualize the results of the study and to inform both the discussion and the recommendations.
SUBHEADING: DIAGNOSIS AND
Major comments
The discussion and the recommendation could benefit from a detailed description of the study setting. While an attempt was made by the authors to describe the Standard of care for people with DR-TB and the study procedures, more detail needs to be added.
RESPONSE: Thank you very much for all your comments and contributions. We have tried as much as possible to make the necessary corrections in the manuscript as uploaded.
Diagnosis and management of DR-TB
IDENTIFICATION FOR SCREENING
If possible, state the diagnostic algorithm that was in use. If LPA is done, is it done on sputum positive deposits or culture deposits? Given that 62% were co-infected with HIV in this study, it might be possible that some people living with HIV produced pauci-bacillary specimens, making preliminary culture a requirement in order to produce visible bands on LPA. However, the authors make reference to use of sputum deposits for LPA in the discussion lines xx.
RESPONSE: This study focus is on the time to treatment irrespective of the diagnostic modality. While Xpert MTB/Rif assay has been decentralized to four centres within the district, the LPA service package was recently added to the National Health Laboratory Services (NHLS); a central laboratory in the Nelson Mandela Academic Hospital resident within the district.
With 90% of the diagnosis made primarily with Xpert/MTB/Rif and only 6.8% diagnosed with the LPA, the difference this would have made if any is negligible. Besides, 42% of the sample were smear (AFB+) positive on the auramine stain making it possible for LPA to be done directly on such sample.
In addition, only the first diagnosis of DR-TB was considered significant as patients were initiated treatment on that basis. Further testing may inform a change or an alteration of regimen and the level of care (decentralized or centralized) but not considered as treatment initiation.
- How are results communicated to patients? (SMSs, calls or they visit the diagnostic facility)
RESPONSE: The patients had to visit the diagnostic facility to get their results which is usually communicated in a very sensitive and ethical manner in the form of post-test counselling, which is followed by pre-treatment counselling and invitation to participate in the study. Line 139-142
- Are patients referred to either decentralized or centralized facilities to start treatment?
RESPONSE: All patients are initiated treatment at designated decentralized DR-TB sites in cases of RR-TB and MDR-TB while cases of pre-XDR and XDR-TB found on further testing (culture and LPA) are referred to the specialist (centralised) centre for proper management. Line 123-125
- Are there DR-TB concillia/committees that have to meet and discuss treatment options for patients?
RESPONSE: No special committee is required as DR-TB is a laboratory diagnosis. What is operational is specially trained staff (Nurses, doctors, data capturers, counsellors) use a programmatic approach to management and take cognizance of special circumstances for referral to specialist centre.
Treatment for DR-TB
Are patients initially started on standardized treatment pending second line DST results to inform individualised treatment? Which group of patients are started on standardized therapy?
RESPONSE: All patients diagnosed with DR-TB using Xpert and LPA1 are started on standardized treatment. Further testing (DST) is usually requested at the specialist centre (centralised site) and on the basis of the result, a salvage (individualized) regimen may be used, but this decision is the exclusive reserve of the DR-TB specialist
Study procedures
May the author chronologically show how data collection unfolded. Perhaps the authors may state that all records of people who were diagnosed with DR-TB were reviewed at diagnostic centers. Data that were collected included date diagnosis, results and sociodemographic details (e.g. names, age and sex) that were used to track patients at decentralized centers and centralized centers. In the next step, they can spell out data that were collected at treatment centres either i) directly from patients (income, education, occupation etc) or ii) from Case report forms and EDRweb (date of starting treatment; age, sex, etc). I would like to believe the dates were collected from records NOT from self-reports by the patients.
RESPONSE: Thank you very much. Incorporated (Line 148-159)
DATA COLLECTION PROCEDURE
STEP 1: Records of all patients from OR Tambo district were reviewed at the diagnostic centres. Data collected included name, date of diagnosis, diagnostic modality, type of resistance, and sociodemographic details such as name, age, and sex. These were used to track the patient at the treatment sites.
STEP 2: At the treatment centres, patients were informed about the research and an……… informed consent form was signed by willing participants following which the patient’s level of income, occupation, education, HIV status, and comorbidities, were ascertained while the name, age, sex were corroborated.
STEP 3: At the treatment centres, the treatment commencement date, age, sex, and type of regimen were collected from the case record and EDRWeb.
Minor comments
Outcome variable: state that it was difference between date of starting treatment and date of diagnosis.
RESPONSE: Done. Thank you so much {Line 169-170)
Lines 322-324: Of note is the finding of no significant difference in treatment initiation time when LPA was used as a diagnostic modality as compared to Xpert in this cohort. This could be explained by the LPA done directly (direct LPA) on smear positive sputum (35, 36) even before MGIT culture results were available.
Since the study focused on interval between diagnosis and starting treatment, diagnostic modality may not have an effect on time to treatment initiation. Differences could have been observed had the study looked at interval from presumption of TB to diagnosis and treatment initiation.
RESPONSE: We agree totally. Special thanks. Necessary correction made.
Lines132-133: Study mentions that part of the inclusion criteria was microbiological diagnosis of DR-TB but in Table 4, there are 15 patients with missing diagnostic modality.
RESPONSE: This is correct! But it is probable that these are the few patients who were diagnosed at private laboratories whose results could not be found or declared missing after efforts made at securing a reprint proved futile. They may also represent patients who were transferred in from other districts or other provinces with referral letters only. Yet, their presence on the EDRWeb could only mean they were confirmed cases of DR-TB whose results could not be objectively verified by the researchers at this time. (Line 302-307)
Define the acronyms used in the table in the legend of Tables. RESPONSE: Included in the tables
Recommendation
RESPONSE: Thank you so much for your inputs. Most have been incorporated into the test. (Line 433-450)
To me one of the key recommendation is that as more and more people are having decentralized treatment, there is need to:
- i)decentralize WHO rapid diagnostics e.g. Xpert MTB/Rif including the Xpert MTB/XDR testing to peripheral/decentralized facilities. This may shorten time from diagnosis to treatment initiation. The Gene Xpert MTB/XDR offers DSTs for isoniazid, rifampicin, FQs and second-line injectables. This assay can be deployed to peripheral areas to provide faster, near patient access to second line DSTs.
- ii)As more people (91% in this study) are now in decentralized DR-TB care, more resources (human and equipment) are required to effectively manage and monitor adverse events of DR-TB at decentralized levels. This should be complemented with mechanisms for early referrals to specialist care at central facilities.
iii) The full benefits of rapid diagnostic technologies may not be realized if contextual factors (health systems delays) to starting treatment for people who are diagnosed with DR-TB are not addressed. Reasons for differences in TTTI in the various health facilities. Is it because of failure to adhere to TB guidelines in some facilities?
RESPONSE: Thank you for your corrections and contributions. Most of them have been incorporated into the test as you will see in the various sections. Line 433-450
It would have been great had the authors given us an idea of the % of people who were switched to effective regimens after CDST results. This is crucial within the context of FQ resistance reported in this study. While shorter time to treatment is important, time to effective treatment is even more important to reduce the time when people are on suboptimal therapies. For this reason, the authors may recommend Xpert XDR tests to be deployed.
RESPONSE: This category would have come from among the patients diagnosed with pre-XDR and XDR-TB and managed at the centralised centre. In all, these represent less than 10% of the total number of patients managed for DR-TB, though they represent a special subset of patients.
Tables: The tables may need a column for the total number of patients in each category. That way it will be easier to deduce how the row percentages were calculated.
Table 3: Check the denominator used to calculate the % of people who attained viral suppression. It seems the denominator used was all patients, including those who were HIV negative.
Line 49: Make it clear that successful outcomes depend on starting effective treatment regimens.
RESPONSE: Incorporated. Thank you.
References: Some parts of the manuscripts are referenced using the APA while others are in Vancouver.
RESPONSE: Corrected using endnote-20 reference manager
Comments on the Quality of English Language
Minor edits of English language are required.
RESPONSE: Done.
Reviewer 2 Report
Major revision required

Major revision required
Author Response
Thank you very much for the hard work of reviewing our paper and making the necessary corrections.
All your recommended corrections have been considered as can be seen in the uploaded file titled "manuscript for review" with track changes left in situ.
Reviewer 3 Report
This is a prospective cohort study to assess the effectiveness of community DR-TB care model to determine time to treatment initiation. It established less than 50% of patients received prompt treatment.
Congratulations to the authors for this well written paper. Some minor suggestions to consider:
1. Inconsistencies in referencing style need fixing
2. What was the reasoning for choosing 2018-2020 only?
3. In 2020, we were dealing with Covid-19 - what influence would this would have had on the results?
4. There is a high unemployment rate in this cohort. Why ? What could be the potential influence to the outcome of the study?
5. Line 304 to 307 - Could the authors discuss this questions please.
Overall a solid paper worth publishing.
Author Response
RESPOINSE TO MDPI REVIEWER No 3.
Dear reviewer,
I was so delighted to read your kind comments and contributions towards our paper. We have made the necessary corrections as you recommended. Thank you for your kind and encouraging comments.
Comments and Suggestions for Authors
This is a prospective cohort study to assess the effectiveness of community DR-TB care model to determine time to treatment initiation. It established less than 50% of patients received prompt treatment.
Congratulations to the authors for this well written paper. Some minor suggestions to consider:
RESPONSE: Thank you kindly for your comments and suggested corrections. Below is our response to your comments.
- Inconsistencies in referencing style need fixing
RESPONSE: The reference manager ENDNOTE 20 was used to correct the referencing inconsistencies.
- What was the reasoning for choosing 2018-2020 only?
RESPONSE: The choice of 2018 - 2020 was informed by the introduction of DECENTRALISED MANAGEMENT of DRTB in the OR Tambo district municipality. In addition, we wanted a period for which we could have outcomes.
- In 2020, we were dealing with Covid-19 - what influence would this would have had on the results?
RESPONSE: The COVID-19 lockdown may have imparted the outcomes negatively due to movement restrictions and business closure as evidenced by WHO report of 2022. This should therefore appear as a limitation to the study. (see Line 423-425)
- There is a high unemployment rate in this cohort. Why? What could be the potential influence to the outcome of the study?
RESPONSE: There is high unemployment in South Africa with rate exceeding 32.8% and higher among youths. Unemployment may lead to poverty which is a major driver of TB. There is an axiom that ‘TB is a disease of poverty”. Upon successful treatment and rehabilitation, it is expected that some patients may be gainfully employed to change their precarious economic circumstance.
- Line 304 to 307 - Could the authors discuss this questions please.
RESPONSE: Yes! And thank you. (Line 329-333)
Overall a solid paper worth publishing.
Submission Date
03 April 2023
Date of this review
23 Jun 2023 10:58:59
Round 2
Reviewer 1 Report
Congratulations to the authors for working so hard to get to this point.
Author Response
Thank you, dear reviewer, for your kind word of encouragement.
Reviewer 2 Report
It is fine

Author Response
Thank you dear reviewer for your helpful comments and corrections. We are grateful.